# Controlling an effector with eye movements: The effect of entangled sensory and motor responsibilities

**John R. Schultz** [1]*, **Andrew B. Slifkin**[2], **Eric M. Schearer**[1]

**1** Mechanical Engineering/Center for Human Machine Systems, Cleveland State University, Cleveland, Ohio, United States of America, **2** Department of Psychology, Cleveland State University, Cleveland, Ohio, United States of America

* j.r.schultz12@vikes.csuohio.edu

**Data Availability Statement:** All relevant data are within the paper and its Supporting information files.

## Abstract

Restoring arm and hand function has been indicated by individuals with tetraplegia as one of the most important factors for regaining independence. The overall goal of our research is to develop assistive technologies that allow individuals with tetraplegia to control functional reaching movements. This study served as an initial step toward our overall goal by assessing the feasibility of using eye movements to control the motion of an effector in an experimental environment. We aimed to understand how additional motor requirements placed on the eyes affected eye-hand coordination during functional reaching. We were particularly interested in how eye fixation error was affected when the sensory and motor functions of the eyes were entangled due to the additional motor responsibility. We recorded participants' eye and hand movements while they reached for targets on a monitor. We presented a cursor at the participant's point of gaze position which can be thought of as being similar to the control of an assistive robot arm. To measure eye fixation error, we used an offline filter to extract eye fixations from the raw eye movement data. We compared the fixations to the locations of the targets presented on the monitor. The results show that not only are humans able to use eye movements to direct the cursor to a desired location (1.04 ± 0.15 cm), but they can do so with error similar to that of the hand (0.84 ± 0.05 cm). In other words, despite the additional motor responsibility placed on the eyes during direct eye-movement control of an effector, the ability to coordinate functional reaching movements was unaffected. The outcomes of this study support the efficacy of using the eyes as a direct command input for controlling movement.

## Introduction

There are approximately 294,000 individuals living with a spinal cord injury (SCI) in the United states. Of these individuals, less than 1% experience neurological recovery and almost 60% of injuries result in tetraplegia, or paralysis of all four limbs [1]. For these individuals, restoring arm and hand function has been indicated as the most important factor for

**Funding:** Schearer, E.M. (PI) received the grant award entitled "Controlling Functional Reaching with Eye and Head Movements of People with High Cervical Spinal Cord Injuries." from the Ohio Department of Higher Education. The award effective dates are 6/1/2020 – 5/31/2022. Grant numbers are not assigned by the Ohio Department of Higher Education. More information can be found here: https://icorpsohio.org/apply/ https://www.ohiohighered.org/grants-rfps The funders had no role in study design, data collection and analysis, decision to publish, or preparation of the manuscript.

**Competing interests:** The authors have declared that no competing interests exist.

maintaining independence [2]. Assistive technologies are a promising solution for regaining lost upper limb function.

One type of assistive technology are robotic arms, which are often fixed to an individual's wheelchair to reach and grasp everyday objects [3–5]. Powered arm supports are similar, however these devices are attached to the limb and can be controlled to move the individual's arm to aid in tasks of everyday living [6]. Another type of assistive technology, wearable robotic exoskeletons, have actuators aligned with the individual's joints to aid in functional movement [7, 8]. Finally, neuroprostheses restore function to an individual's limb by electrically stimulating the muscles through Functional Electrical Stimulation (FES) [9]. These assistive technologies have shown promise in restoring reaching function to individuals with SCI. However, a reliable method for sending motor commands from the user to the assistive device through a human machine interface (HMI) remains elusive.

Individuals with C4-level spinal cord injuries and above are paralyzed from the neck down, meaning the only available command signals that can be used as inputs to the HMI are located above the neck and shoulders. Previous studies have used surface electromyography (EMG) to trigger actions from the assistive device in response to neck muscle movements [10, 11]. Additional interfaces include sip-and-puff systems [12], tooth clicking [13], magnetic tongue switches [14], and speech recognition [15]. While effective for simpler tasks, these command signals generally lack the throughput and sensitivity necessary for complex motor tasks such as unconstrained reaching [16].

The richest source of command signals has its origin in the brain. Electroencephalography (EEG) involves placing electrodes on the surface of the scalp to record brain activity. Recording signals through the scalp and cranium leads to low spatial resolution and a low signal to noise ratio [17, 18], severely limiting the number of commands that can be decoded for controlling complex reaching motions. Conversely, electrocorticography (ECoG) maintains a higher spatial resolution and a higher signal to noise ratio due to the implantation of electrodes directly on the surface of the brain. Recent studies have used ECoG to record command signals for controlling reaching with an assistive device [19, 20]. While promising, ECoG involves highly invasive surgical procedures, requires extensive recovery times, and is accompanied by high clinical and post-clinical costs [21].

In contrast, eye movements are naturally available, directly observable, and highly correlated with upper-extremity motor function [22–24], and as such are candidate input sources for controlling reaching. However, eye movements contain rapid changes in acceleration, and humans do not maintain focus on one object of interest for the duration of a task [25, 26]. Therefore, it may be difficult for people to control the position of an assistive device directly. Some studies have supplemented eye-tracking with other sensors to reduce the cognitive burden on the user. Recent studies used computer vision and object recognition to aid in target selection [27–29], as well as finite state machines to assist in intention prediction [30]. Combining inputs in this way is one approach that has shown promise in controlling assistive devices. In the previously mentioned studies, eye tracking was used primarily to select targets that were then reached for or acquired by the assistive device through perception and planning. In the current work, we are interested in investigating using eye movements to directly inform effector movement as an alternative approach for controlling assistive devices.

Because eye movements contain rapid changes in acceleration and are not always directed smoothly toward a target, it is necessary to apply filters that extract important features which characterize the movement. These filters are based on the natural behavior of the eyes. In order to view an object with a high level of detail, the eye must be rotated such that the desired object can be viewed by the fovea, the area of the eye with the highest visual acuity. These rotations are executed through quick, ballistic movements called saccades. The periods of time

between successive saccades are called fixations, which indicate where a person is directing their focus at a specific moment in time [31]. During natural functional reaching, the eyes generally fixate around the desired object, but they do not necessarily maintain constant focus on the target throughout the task [25, 26]. Positional errors in reaching are corrected through visual feedback from the eyes and proprioceptive feedback from the arm and hand [32]. For individuals with an SCI, proprioceptive feedback from the arm and hand is unavailable.

When controlling an assistive device with eye movements, the device serves as a replacement for the function of the arm and hand. In this case, the eyes must now provide the command input that determines the position of the end effector of the assistive device as well as observe errors in the end effector position. In this sense, the eyes serve as both an actuator to guide the effector as well as a sensor to observe positional errors. This leads to a situation in which the responsibilities of the eyes are *entangled*, meaning, the execution of the eyes' sensory and motor roles affect each other. For example, in order for the eyes to observe the position of the device's end effector, the eyes must rotate, which then moves the end effector further. If there is a large error between the end effector position and where the human is looking, a positive feedback loop can be generated, which causes the end effector to drift. The same phenomenon can occur when using eye movements to control a mouse cursor on a monitor [33]. The goal of this study was to determine if humans can use eye movements to control a cursor to target locations despite the entangled sensory and motor responsibilities that this method introduces. To our knowledge, the effect which the additional motor responsibility on the eyes and the resulting entangled sensory and motor functions have on humans' ability to control an effector with eye movements remains unknown. However, there is some evidence to suggest that presenting visual feedback to the user improves performance and user experience [33, 34].

We undertook this study to assess the feasibility of using eye movements to directly control an effector. Specifically, we addressed the following research questions: First, how accurately can humans direct an effector to a desired location with their eye movements when the eyes' sensory and motor responsibilities are entangled? Second, how does the eye fixation error compare to hand endpoint error when there is no eye-function entanglement? The results of this study will inform continued research on direct eye-movement control of assistive technologies.

## Materials and methods

In this study, we aimed to determine if humans can direct an effector to desired target locations using eye movements. More specifically, we aimed to understand the effect of the eyes' entangled sensory and motor responsibilities on eye fixation error during functional reaching. We collected eye and hand movement data while participants performed reaches during different experimental conditions and compared the eye fixation error between them. Conditions varied by whether the participant was instructed to reach for, or just look at targets displayed on a monitor, as well as the inclusion or omission of a cursor. The cursor provided feedback for the eye tracker's estimation of the participant's point-of-gaze.

### Participant information

A total of 7 individuals (5 male, 2 female) participated in this study. They ranged between the ages of 23 and 30. None of the participants had any neurological or visual disorders, and all participants had normal or corrected-to-normal vision. Informed written consent was obtained for each participant according to the testing protocols approved by the Institutional Review Board at Cleveland State University (IRB-FY2019–93). Participants were informed of the experimental protocol and agreed to all testing in writing. Participants were also informed

and provided written consent for their data to be de-identified and used in scientific publications and research. Prior to this study, none of the participants were familiar with eye-tracking systems nor were they expert users. Each testing session lasted approximately one hour, and participation was voluntary.

## Experimental conditions

Each participant performed the experiment under four conditions, in a randomized order:

1. Eye-Alone

2. Eye-Hand

3. Eye-Alone with Cursor

4. Eye-Hand with Cursor

In all of the conditions, participants were instructed to select targets displayed on the monitor by moving an effector to each target. For this study, we refer to the *effector* as the element being moved to the target, and we refer to the *actuator* as the motor command that moves the effector. Errors in the effector position are observed by the *sensor*. For example, in the case of controlling an assistive-robotic arm with eye movements, the effector is the end effector (usually a gripper) of the robot, the actuator is the muscles of the eyes, and the sensor is the visual feedback received by the eyes. Of course, the motors of the robot are what physically move the robot's end effector, but in this study, we use the term "actuator" to refer to the command input that determines the desired location of the effector. Likewise, the robot may contain internal sensors such as potentiometers and accelerometers, but in this study, we use the term "sensor" to refer to the mechanism by which the human observes positional errors in the effector. While these definitions may be oversimplified, defining terminology in this way should help to clarify the differences between experimental conditions (Fig 1).

In the 'Eye-Alone' condition, participants were instructed to look from the center of the starting target to the center of the task target (see Visual stimuli), and to maintain focus on the task target while it was displayed. No special instructions were given about how to move their eyes. In this condition, the position of the point-of-gaze was the effector and the muscles of the eyes were the actuator. When foveating a target, the change in the visual field as well as eye-muscle proprioception provide feedback to help direct the point-of-gaze to the desired location. Therefore, we simply label the eyes as the sensor in this condition. This condition was meant to provide insight into how humans foveate objects without a movement response from an assistive device.

The 'Eye-Hand' condition was similar to the 'Eye-Alone' condition, except that the hand component was introduced. Participants were instructed to move their index finger from the center of the starting target to the center of the task target and to maintain hand position on the target while it was displayed. No special instructions were given about how to move their eyes or hand. In this condition, the position of the finger was the effector and the muscles of the arm and shoulder were the actuator. Positional errors in the effector position were observed both through visual feedback by the eyes and through proprioceptive feedback by the muscles of the hand, arm, and shoulder. This condition was meant to provide insight into how humans look at and reach for objects without a movement response from an assistive device.

In the 'Eye-Alone with Cursor' condition, a cross-hair cursor was displayed at the eye tracker's estimation for the participant's point-of-gaze. In effect, the cursor provided visual feedback for where the participant was looking on the monitor and could be thought of as being similar to the control of the endpoint of an assistive device. Participants were instructed to move the

**Fig 1. Experimental conditions.** Summary of important characteristics for each experimental condition. Conditions were characterized by different combinations of goal, effector, actuator, and sensor. Each participant performed all experimental conditions in a randomized order.

cursor from the center of the starting target to the center of the task target, and to maintain cursor position on the task target while it was displayed. No special instructions were given about how to move their eyes or hand. In this condition, the cursor was the effector. The eyes served as both the actuator to move the effector as well as the sensor to observe errors in the effector position, thus the sensory function and concurrent motor responsibility of the eyes were *entangled*. In the context of an assistive system, this condition was meant to represent the case of direct gaze control of an assistive device.

The 'Eye-Hand with Cursor' condition was similar to the 'Eye-Hand' condition, except the cursor was displayed at the participant's point-of-gaze for the duration of the condition. Participants were instructed to move both the cursor and their index finger from the center of the starting target to the center of the task target, and to maintain both cursor and hand position on the center of the target while it was displayed. No special instructions were given about how to move their eyes or hand. In this condition, the existing proprioceptive feedback from the arm and hand was not altered, but the eye's source of feedback was augmented by the inclusion of the cursor. Here, the cursor and the hand were both effectors, with their corresponding actuators and sensors being the eyes and the muscles of the arm and hand, respectively. In the context of an assistive system, this condition might represent teleoperation, where the human reaches for objects remotely, and triggers a movement response from an assistive device.

## Experimental setup and data acquisition

For the duration of the experiment, the participant was seated behind a table. On the table was a 55" Samsung monitor (60 Hz refresh rate), within arm's reach of the participant (Fig 2). Targets were displayed on the monitor and the participant was instructed to look at and reach for the targets (see Visual stimuli). A 6" tall foam arm rest was placed on the table between the

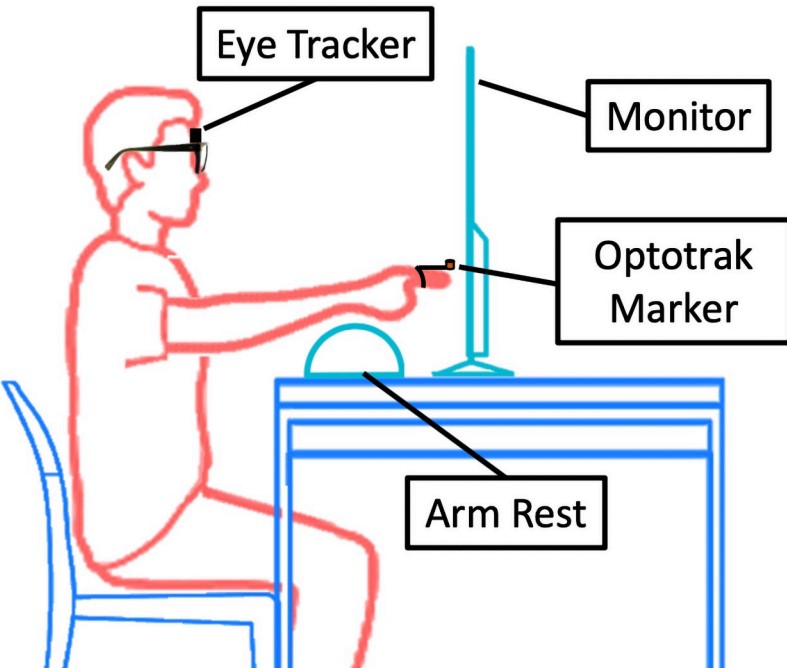

**Fig 2. Experimental setup.** Each participant was seated at a table within reach of a monitor, which displayed task targets. Eye movements were recorded with a commercial head-mounted eye tracker. A motion capture marker was affixed to the participant's finger to record hand movements. The participant was permitted to use the arm rest between reaches to reduce fatigue.

participant and the monitor. The participant was permitted to rest their arm on the arm rest between reaches to reduce the effects of fatigue during data collection. Participants were positioned in front of the monitor such that they could comfortably reach all of the target locations, and their forward gaze fell approximately on the center of the screen. All experiment logic was programmed in the MATLAB Simulink environment (Mathworks; Natick, MA) and executed on a microLab box real-time target computer (dSPACE; Paderborn, Germany). Eye movements were recorded with the ETL-600 head-mounted eye-tracking system (iSCAN; Woburn, MA) with a sampling frequency of 240 Hz. This system consisted of two infrared cameras that monitored the position of the pupils. It utilized a 6 dof magnetic head position sensor to allow free head movement. The point-of-gaze was computed through iSCAN's 5-point calibration procedure and returned as $x$ and $y$ screen coordinates. When collecting eye-tracking data, the head may be constrained to prevent head motion. However, this study was designed to investigate whether eye-movements are a good candidate input signal for informing reaching motions in the context of gaze-controlled assistive devices. Under those circumstances, the head would not be constrained. Therefore we wanted to allow free head motion during data collection. Additionally, in this study, we were only interested in the participant's point-of-gaze on the monitor in relation to the targets. Further, the eye-tracking system accounts for variations in the head orientation in the computation of the point-of-gaze. Finger position was recorded using the Optotrak 3D Investigator (Northern Digital; Waterloo, Canada), sampled at 1000 Hz and accurate to 0.4 mm. A marker was affixed to a stylus and attached to the participant's index finger to monitor hand movement. The coordinate systems of the monitor and the Optotrak were aligned for ease of data analysis. Signals from the eye-tracker and the motion capture system were sent to the microLab box, which facilitated clock synchronization,

experimental logic, and data logging. Raw experimental data can be found in the attached S1 File.

## Visual stimuli

Each participant was instructed to perform a series of reaches, in which a starting target was displayed in the bottom center of the screen, and 6 task targets were displayed separately, in a randomized order (Fig 3). During initial tests of the experimental protocol, we found that if the starting target was displayed in the center of the screen, the participant's arm would rapidly fatigue. Thus, we moved the starting target to the bottom of the screen and included an arm rest to reduce arm fatigue. All visual stimuli were displayed on the monitor using the OpenGL graphics framework. White targets on a black background were selected to reduce eye strain. Target stimuli were presented as a combination bulls-eye/cross-hair with a 2.5 cm diameter, which has been shown to attract participant attention and reduce gaze dispersion [35].

At the beginning of an experimental condition, only the starting target was displayed. After a delay, one of the task targets appeared and the participant looked at and/or reached for (depending on the experimental condition) the task target. After another delay, the task target disappeared and the participant returned their gaze and/or their hand back to the starting target. Delays were chosen randomly between 2 and 5.5 s to prevent participants from anticipating target movement. Looking and/or reaching from the starting target to the task target and back to the starting target was considered one trial. Participants performed approximately 8 trials per target location, depending on the randomization, for each of the four experimental conditions.

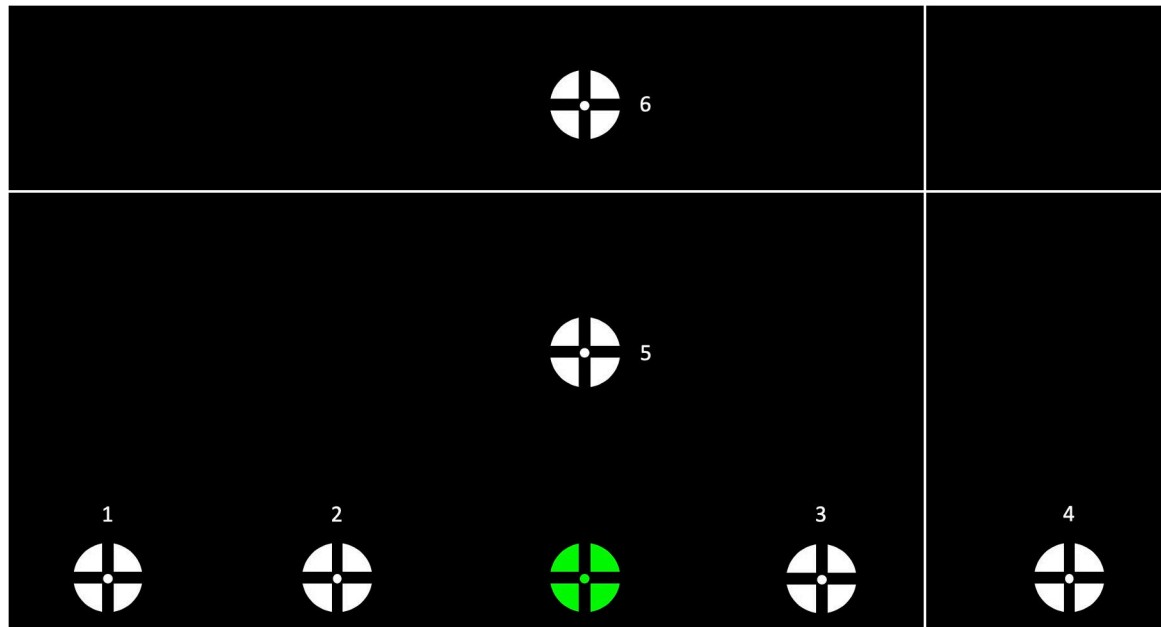

**Fig 3. Visual stimuli target array.** Distances from the green starting target centers to the center of each white task target were as follows: 1) 24.2 cm, 2) 12.1 cm, 3) 12.1 cm, 4) 24.2 cm, 5) 13.0 cm, and 6) 26.0 cm. Here, the starting target is shown in green, however during the experiment, all targets were white. This figure depicts the locations that the targets would appear on the monitor, as well as an example position of the cross-hair cursor. During the experiment, only one target would be present on the screen at a time.

## Analysis

To extract saccades (eye movements) and fixations (stable periods between saccades) from the raw eye data, we applied an offline fixation filter based on the methods by Olsson [36]. The first step was to interpolate any missing data due to blinks or dropped frames. When a participant blinked, the system returned zeros for the point-of-gaze coordinates and those values were flagged. The second step was to define a sliding window size within which large changes of the signal mean were detected. Typical fixations last between 150 and 600ms [31], rarely lasting less than 100ms [37]. The sliding window should be large enough to adequately detect changes in the signal mean, but not too long to span multiple fixations. Thus, we defined our sliding window as 80ms. The third step was to find the 'peaks' within the sliding window. For this step, we used the MATLAB function `findchangepts()`, which employs a parametric global method for determining changes in the signal mean. The fourth step was to remove peaks that occur too close together in the time domain. If more than one peak occurred within the sliding window, only the peak with the highest magnitude was recorded. The final step was to estimate the spatial positions of the fixations using the median and merge fixations that occurred within a specified radius. In this case, we used 1.1cm as the merge threshold. Once fixations were detected, we computed the eye fixation error as the Euclidean distance between the center of the task target and the extracted eye fixation position, and then computed the mean eye fixation error across all targets for each participant within a given condition. We then took the average across all participant means as the group-mean eye fixation error for that condition. The normality assumption was verified for within-participant as well as group eye fixation errors. For clarification, the computation of the eye fixation error was identical in the 'Cursor' and 'No Cursor' conditions. The only difference between conditions was that the cursor was displayed to provide augmented visual feedback to the participants.

In addition to detecting fixations from eye movement data, we also extracted the hand movement endpoints from each hand trajectory, which estimate the point in space where the movement is terminated. Specifically, the endpoint was identified as the point in the movement trajectory where movement velocity dropped below a threshold of 10 cm/s [38]. If there were multiple instances where the hand movement velocity crossed the threshold within a one second time frame, then the endpoint was based on an average of those positions. We computed the hand endpoint error as the Euclidean distance between the center of the task target and the hand movement endpoint, and found the mean hand endpoint error across all targets for each participant within a given condition. We then took the average across all participant means as the group-mean hand endpoint error for that condition. The normality assumption was verified for within-participant as well as group hand endpoint errors.

We compared the group-mean eye fixation errors between experimental conditions to understand how eye fixations were affected by the entangled sensory and motor responsibilities of the eyes depending on the condition. Statistical analyses were performed using a two-way, effector by cursor repeated-measures ANOVA with the following factors:

1. Effector: Eye-Alone/Eye-Hand

2. Feedback: No Cursor/Cursor

In the current study, we were interested in how the eye fixation error when the visual feedback was augmented by the inclusion of the cursor compared to the hand endpoint error when the eyes' normal feedback modality was present. To do this, we compared the group-mean eye fixation error in the 'Eye-Alone with Cursor' condition to the group-mean hand endpoint error in the 'Eye-Hand' condition. This can be thought of as similar to comparing the control of an assistive device with eye movements to natural reaching, albeit with much

less complexity. Comparing these two conditions can give us an indication of the effectiveness of replacing lost arm and hand function with a gaze-controlled assistive device. Statistical analyses were performed using a paired-samples equivalence test. Statistical analyses were performed using both SPSS statistical analysis software (IBM; Armonk, NY) and Minitab statistical software (Minitab; State College, PA).

## Results

### Fixation estimation and hand movement detection

Using an offline fixation filter, we computed the fixation positions from the raw eye movement data. Similarly, using a velocity threshold, we identified the hand movement endpoints. The typical temporal sequence of events consisted of the appearance of the task target, followed by the eye fixation, and finally the hand reaching the target location (Fig 4). This is consistent with observations made in other studies [38, 39].

Raw eye movement data contains high signal variability due to the rapid, saccadic nature of the eye movements combined with the fact that the eyes do not necessarily focus on the object of interest throughout the duration of reaching tasks [22–24, 31, 36]. The plot in Fig 5A illustrates the variable nature of the eye movement trajectories, yet fixations are accurately estimated. In contrast, the hand movement trajectories are smoother as the hand cannot

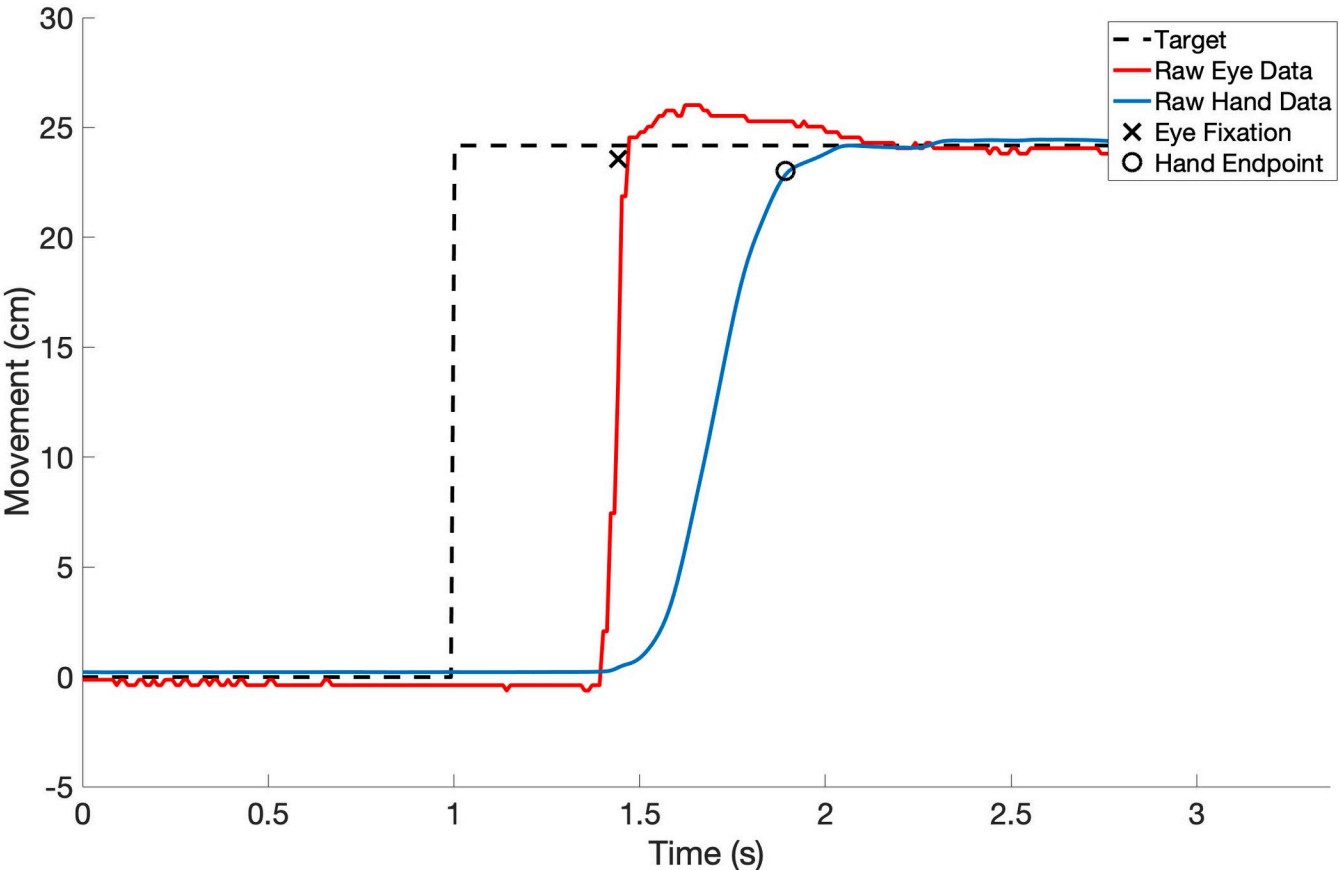

**Fig 4. Temporal sequence of events.** One trial is defined as movement from the starting target, to the task target, and back to the starting target. Just the movement in the horizontal direction from the starting target to the task target is shown here for one representative trial. A typical trial sequence consisted first of the task target appearance, followed by the eye fixation, followed by the hand endpoint.

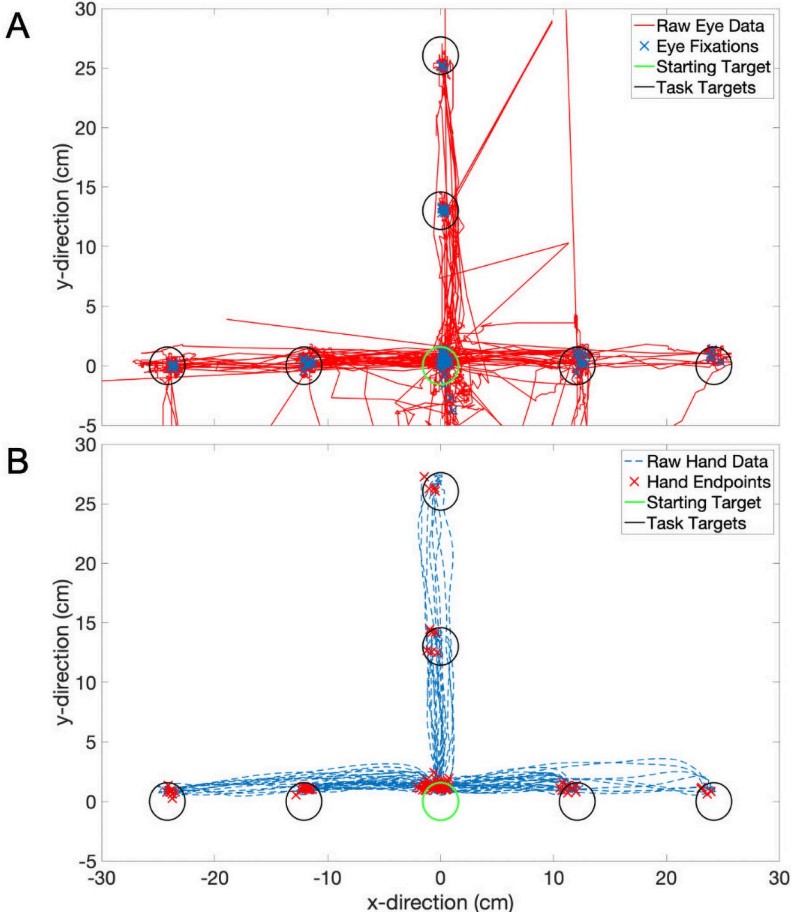

**Fig 5. Eye fixations and hand endpoints.** Representative participant data collection session for the 'Eye-Hand with Cursor' condition. Forty trials in the session are shown here. A) Raw eye movement data with eye fixations (blue **X's**) for all trials. B) Raw hand data with marked hand positions at the end of the hand movement (red **X's**) for all trials.

accelerate as quickly as the eyes (Fig 5B). Another possible reason for this difference in signal variability is the difference in sampling rates between the eye-tracker and the motion capture system. However, we have shown that an offline filter can be used to accurately estimate fixation positions in eye movement data despite the inherent signal variability. Raw subject data can be found in S1 File.

## Eye fixation and hand endpoint error

As described in Analysis, we performed a two-way effector ('Eye-Alone', 'Eye-Hand') by cursor ('Cursor', 'No Cursor') repeated-measures ANOVA. The eye fixation and hand endpoint group-mean error values in each experimental condition are shown in Fig 6. The first result we report is the effect of the inclusion of the hand in the task on the eye fixation error. While descriptive statistics revealed that participants' mean eye fixation error was slightly lower when using their eyes alone (1.71 ± 0.35 cm) compared to when movements were made with both their eyes and hand (1.84 ± 0.32 cm), the results of the ANOVA revealed that there was no significant main effect of the type of effector on eye fixation error (F(1,6) = 1.814, $p > .05$, $\eta_p^2 = .232$). In other words, whether or not the participant reached with their hand or just

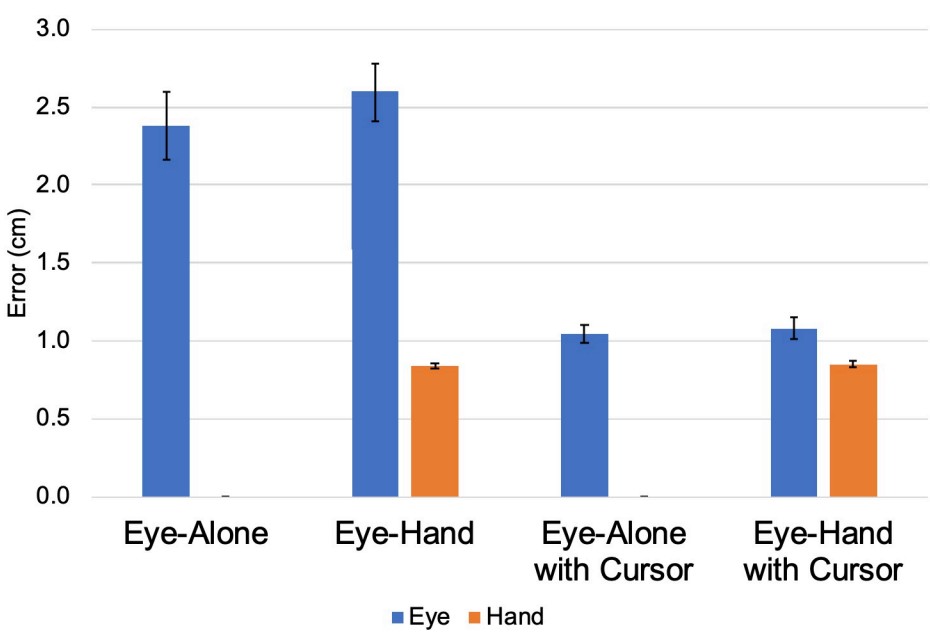

**Fig 6. Eye fixation and hand endpoint error.** Group-mean eye fixation error and hand endpoint error for each condition including standard error of the mean.

looked at targets did not affect their eye fixation error. For clarity, these values come from averaging the 'Eye-Alone' conditions ('Eye-Alone' and 'Eye-Alone with Cursor') and the 'Eye-Hand' conditions ('Eye-Hand' and Eye-Hand with Cursor), respectively.

The second result we report is the effect of the inclusion of the cursor in the task on the eye fixation error. According to the ANOVA, there was a significant main effect of the cursor on participants' eye fixation error ($F(1,6) = 11.983$, $p < .05$, $\eta_p^2 = .666$) such that participants' mean eye fixation errors were lower when the cursor was displayed ($1.06 \pm 0.17$ cm) compared to when the cursor was omitted ($2.49 \pm 0.53$ cm). In other words, when the cursor was presented, the error between the effector position and the target was reduced compared to when the cursor was omitted. For clarity, these values come from averaging the 'with cursor' conditions ('Eye-Alone with Cursor' and 'Eye-Hand with Cursor) and the 'without cursor' conditions ('Eye-Alone' and 'Eye-Hand'), respectively.

The final result we report is the equivalence between the eye fixation error in the 'Eye-Alone with Cursor' condition and the hand endpoint error in the 'Eye-Hand' condition. We used a paired-samples equivalence test to compare the group-means from those conditions. We defined the equivalence interval for the difference in means to be ± 1cm. This equivalence interval corresponded with the diameter of the stylus attached to the finger. Because the trial instructions were simply to touch the target center with the tip of the stylus, we assumed the error of the reach could not be less than the diameter of the stylus, as reflected by the equivalence interval. While the mean hand endpoint error in the 'Eye-Hand' condition ($0.84 \pm 0.05$ cm) was slightly lower than the mean eye fixation error in the 'Eye-Alone with Cursor' condition ($1.04 \pm 0.15$ cm), the results of the equivalence test revealed the means were equivalent within ± 1cm ($t(6) = 7.1$, $p < .05$ and $t(6) = -4.71$, $p < .05$). In other words, participants were able to direct an eye-driven cursor to the target location with error like that of the hand during reaching with no cursor presented.

In summary, the overall quality of participants' eye-movement performance was similar whether or not concurrent hand reaches were made, while the addition of an eye-driven cursor resulted in a lower mean effector error across all participants. Moreover, participants' lowered eye fixation error in the 'Eye-Alone with Cursor' condition was comparable to the hand end-point error in the 'Eye-Hand' condition (Fig 6).

## Discussion

Previous studies have investigated how the functions of the eyes and hand interact to perform reaching tasks [22–26, 38–44]. In the current study, we aimed to understand if humans are able to perform similar reaching tasks when the normal sensory responsibility of the eyes is entangled with an atypical motor responsibility. We induced this situation by introducing an additional effector (cursor) that the participants were required to move to a target with their eye movements. We measured the performance of participants' reaches using various offline methods as described in Analysis. The main finding of the current study is that not only are humans able to use eye movements to direct a cursor to a desired location, but they can do so with error similar to that of the hand during reaching without augmented visual feedback. The outcome of this study suggests that humans are able to control effector movement while simultaneously processing visual feedback for the effector position, and that gaze-control may be an effective replacement modality for lost arm and hand function. These results highlight the potential for eye-controlled motion of assistive technologies.

### Fixation estimation

Through the use of an offline fixation filter, we computed the participants' fixation positions from their eye movement trajectory data. Despite the observed high signal variability, we were able to estimate fixation positions (Fig 5A). In the 'Eye-Alone' condition, the participants' existing feedback mechanisms were not augmented, and thus the mean eye fixation error from this condition (2.38 ± 0.22 cm) is a good indication of the fixation filter's performance under normal conditions.

### Directing the cursor with eye movements

As discussed in (Eye fixation and hand endpoint error), a repeated-measures ANOVA was used to elucidate the effects of both the use of the hand and the inclusion or omission of the cursor on participants' ability to look at and/or reach for targets. The ANOVA revealed that when participants were instructed to select targets with their hand, there was no effect on their eye fixation error compared to when they were instructed to select targets with their eyes. This is an encouraging finding, particularly for individuals with a spinal cord injury, because it suggests that humans' ability to foveate objects during a reaching task should not be hindered by the removal of the arm and hand. In addition, the ANOVA revealed that the inclusion of the cursor significantly decreased participants' eye fixation error. This finding may suggest that providing point-of-gaze feedback improves the ability of humans to foveate objects. However, it is important to remember that the eye fixation error is a measure of the difference between the cursor and the desired target, where the cursor appears at the location that the eye-tracking system estimates the user to be focusing their gaze. This means that in some cases, the participant may be looking at the center of the desired target, but the point-of-gaze position contains slight errors, resulting in a shift in the cursor position. In the conditions where the cursor was not displayed, the participant had no knowledge of any potential discrepancy between where they were looking and the position of the cursor. Therefore, the decrease in eye fixation error from the 'without cursor' to the 'with cursor' conditions in Fig 6 can be understood as the

participants' ability to correct for errors in the position of the effector. This finding is encouraging because it suggests that if there are slight errors present in the position of the end effector of an assistive device comparable in magnitude to those experienced in this study, humans may be able to correct these errors using eye movements.

The results here seem to suggest that in an experimental environment, humans can accurately direct an effector to a desired location with their eye movements, despite both the presence of small errors in effector position as well as the additional actuation responsibility placed on the eyes. This, however, may be dependent upon the magnitude of the eye-tracking error. The research question regarding the comparison to hand performance without augmented visual feedback is answered through an analysis of the eye fixation error in the 'Eye-Alone with Cursor' condition and the hand endpoint error in the 'Eye-Hand' condition (Fig 6). The paired-samples equivalence test revealed that the means were equivalent within ± 1cm. This means that when the cursor feedback was presented, participants directed the cursor to the target location with error similar to their hand during reaching when the cursor was omitted. This finding is encouraging because it suggests that gaze-control may be an effective replacement modality for lost arm and hand function.

## Limitations, implications and future directions

There are a number of limitations to the current study design that are worth discussing. The first limitation is the small sample size. We performed a power and sample size test and found that with the observed difference in means, we had an observed power of 75% and 83% for the ANOVA and equivalence tests, respectively. To achieve a power level of at least 90% in both measures, we would need a sample size of at least 11 participants. Another potential source of error were inconsistent number of reaches per target. On occasion, the participant would fail to notice the appearance of a target, particularly the outer targets, and would not look at or reach for the target. We believe this is due to the fact that participants relied on their peripheral vision to alert them when a target had appeared. This resulted in some targets being missed and an inconsistent number of data points per target. Finally, there may have been slight differences in the distance participants were from the monitor. This may have contributed to differences in eye fixation error. However, each participant performed all of the experimental conditions in the same position. Therefore, between-participant differences may be present, but within subject differences should be relatively small. Future studies may want to consider controlling for this potential source of error.

The goal of this study was not to solve all of the issues facing real-time, gaze-controlled assistive devices, but rather, to investigate the effects of augmented visual feedback from introducing an additional effector and the resulting entangled sensory and motor responsibilities placed on the eyes. It may be important, however, to discuss the potential challenges to controlling real assistive technologies with eye movements in light of the findings of this study. We have shown that humans are able to correct for errors in effector position with their eye movements. However, this study was conducted in a controlled, laboratory environment in a two-dimensional plane. In real-world, unconstrained reaching with an assistive device, the errors present in the end effector position may be higher in magnitude than those experienced in this study. The error will likely be compounded by additional sources of error such as the estimation of the depth component of the user's point-of-gaze, as well as end-effector positional error present in the assistive device itself. Additionally, controlling the endpoint of an assistive robot arm will introduce noise and movement delays that are not present in the cursor displayed on the monitor in the present experiment. It is currently unclear if humans will

be able to correct for these potentially greater sources of error, but the results of this study show the potential for the correction of positional errors with small magnitude.

Another common challenge associated with direct gaze-control of movement is the 'Midas Touch' problem [44]. This refers to a system's inability to differentiate between eye movements intended to initiate movement and those intended simply for perception. When controlling a physical assistive device, the Midas Touch problem must be addressed. Common solutions are to introduce an additional modality such as a blink, a speech command, or a dwell time. In the current work, we focused only on the eye-movements directly responsible for effector control. The results of this study suggest that humans are able to direct an effector to a desired target location and can cope with the Midas Touch problem if their focus is directed on the task at hand, albeit for a limited amount of time.

In the current study, we have shown that humans are able to use their eye movements to direct an effector to a desired target location despite the additional motor responsibility placed on the eyes. Moreover, the current results were based on data collected during a single experimental session, which lasted about an hour. Thus, with minimal task experience and training, the addition of point-of-gaze positional feedback resulted in performance gains similar to the performance of the hand during reaching with unaltered visual feedback. Although many challenges still remain, the results of this study suggest that eye movements may be used to directly control an effector to desired locations, and validates continued research toward extending these principles to controlling physical assistive technologies.

## Supporting information

**S1 File. Participant raw rata.** This zip file contains the raw gaze and hand endpoint positional data for each participant and each experimental condition.
(ZIP)

## Acknowledgments

The authors would like to thank the staff at iSCAN Inc. (Woburn, MA) for hardware and software support. The authors would also like to thank Atul Gopal (National Brain Research Centre, Manesar, Haryana, India) for eye-hand coordination expertise and for providing test data in the early stages of the study.

## Author Contributions

**Conceptualization:** John R. Schultz, Andrew B. Slifkin, Eric M. Schearer.

**Data curation:** John R. Schultz.

**Formal analysis:** John R. Schultz.

**Funding acquisition:** Eric M. Schearer.

**Investigation:** John R. Schultz, Andrew B. Slifkin, Eric M. Schearer.

**Methodology:** John R. Schultz, Andrew B. Slifkin, Eric M. Schearer.

**Project administration:** Eric M. Schearer.

**Resources:** John R. Schultz, Eric M. Schearer.

**Software:** John R. Schultz.

**Supervision:** Andrew B. Slifkin, Eric M. Schearer.

**Validation:** John R. Schultz.

**Visualization:** John R. Schultz.

**Writing – original draft:** John R. Schultz.

**Writing – review & editing:** John R. Schultz, Andrew B. Slifkin, Eric M. Schearer.

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
