## [Decision Letter · Decision Letter 0]

6 Oct 2021

PONE-D-21-26505Controlling an Effector with Eye Movements: The Effect of Entangled Sensory and Motor ResponsibilitiesPLOS ONE

Dear Dr. Schultz,

Thank you for submitting your manuscript to PLOS ONE. After careful consideration, we feel that it has merit but does not fully meet PLOS ONE’s publication criteria as it currently stands. Therefore, we invite you to submit a revised version of the manuscript that addresses the points raised during the review process.

We look forward to receiving your revised manuscript.

Kind regards,

Markus Lappe

Academic Editor

PLOS ONE

Journal Requirements:

2. Please amend your Methods section to state that participants provided informed consent, and please state the type of consent (i.e., written, verbal, etc.).

“Schearer, E.M. (PI) received the grant award entitled “Controlling Functional Reaching with Eye and Head Movements of People with High Cervical Spinal Cord Injuries.” from the Ohio Department of Higher Education. The award effective dates are 6/1/2020 – 5/31/2022. Grant numbers are not assigned by the Ohio Department of Higher Education. More information can be found here:

https://icorpsohio.org/apply/

https://www.ohiohighered.org/grants-rfps”

Reviewers' comments:

Reviewer's Responses to Questions

**Comments to the Author**

1. Is the manuscript technically sound, and do the data support the conclusions?

Reviewer #1: Yes

Reviewer #2: Partly

2. Has the statistical analysis been performed appropriately and rigorously? 

Reviewer #1: No

Reviewer #2: Yes

3. Have the authors made all data underlying the findings in their manuscript fully available?

Reviewer #1: Yes

Reviewer #2: Yes

4. Is the manuscript presented in an intelligible fashion and written in standard English?

Reviewer #1: Yes

Reviewer #2: Yes

5. Review Comments to the Author

Reviewer #1: The authors report an experiment on the usability of eye movements to control the motion of an effector, i.e. a tool that is moved toward a target object.

The background is that people with movement impairments, for example as a result of a spinal cord injury, lose movement control over their limbs. Restoring hand and arm function is essential for people to be able to perform everyday tasks and regain independence. This can be achieved, for example, with the help of robotic arm prostheses. One of the ways to control these aids can be eye movements, as they are closely connected to our actions. For example, people usually direct their gaze to the objects they want to grasp next.

In this study, the authors investigated whether additional motor demands on the eyes, which normally serve to assess the accuracy of e.g. hand movements, affect eye-hand coordination. The authors investigated whether, when controlling an effector by means of eye movements, the combination of the usual sensory and the additional motor demands leads to a decrease in fixation accuracy. The main question of the study was whether, when controlling a cursor with the eyes, i.e. combining motor and sensory demands on the eye, fixation accuracy is lower than the accuracy with which people reach for objects by hand under natural conditions.

Answering this question is relevant when evaluating if eye movements are a suitable means to control assistive devices. The present study is thus timely and important. The finding that the participants in this study were capable of directing an effector to a target object via eye movements as accurately as they reached for a target object with their hand highlights that the visual system can handle the combined motor and sensory demands. This study is thus an encouraging starting point for further research on the role of eye movements in the control of prostheses and other assistive devices. Although in my opinion the experimental design is appropriate for the research question and the conclusions drawn are sound, I nevertheless have come critical comments, especially on the statistical analysis.

Statistical Analysis:

First, I noticed that comparisons between conditions and factor levels are initially made on the basis of descriptive values. It is not clear to me why no statistical test is carried out before conclusions are drawn about possible differences between conditions or different factor levels (p. 7, ll. 260-278), especially since not all of those assumptions are subsequently validated by the ANOVA (e.g. p. 7, ll. 260-264; pp. 7-8, ll. 275-279). Moreover, later on, when the results of the ANOVA are reported, the descriptive values described in the beginning of the results section are repeated (e.g. p.7, l. 262 and p. 8, l. 286). This seems redundant.

Second, I stumbled over the phrase “to verify these interactions” (p. 7, ll. 274) for two reasons. First, the previously reported descriptive values were not indicative of an interaction. Second, no interaction is verified in the following. Instead, only the main effects of the ANOVA are reported while the interaction of cursor and effector is not mentioned at all.

Third, the main result of the study, namely that “participants were able to direct an eye-driven cursor to the target location with accuracy like that of the hand during reaching with no cursor presented” is based on a non-significant t-test. It is important to distinguish between a non-significant finding, implying that there is no substantial difference in hand-reaching and fixation error, and actual evidence of equal error magnitude. Especially given the small sample size of this study, I would be cautious about interpreting a non-significant result. Since the finding that there is no significant difference in accuracy is the central finding of the study, I would suggest performing an additional Bayesian t-test on the data that can actually provide evidence for H0 and thus equal accuracy/error magnitude. If it proves difficult to determine a prior, the test could still be run over a range of prior widths. In addition, I suggest the authors acknowledge the small sample size in the section “Implications and future directions” where they discuss the limitations of their current work.

Fourth, the authors have specified the degrees of freedom of the t-test as 7 (p. 8, l. 295). Since the sample consists of 7 participants, there seems to be an error here.

Fifth, during reading I noticed that it was easy to get confused with the terms fixation accuracy, fixation error, hand endpoint error, hand movement endpoint error, hand error and hand-reaching error. On p. 8 ll. 292-293 the authors wrote that they compared the hand endpoint error with the mean eye fixation accuracy while they probably meant that they compared the fixation error with the hand endpoint error. I would recommend using a more consistent terminology. This would further improve the readability and clarity.

Figures:

Figure caption 1: Even though it is already mentioned in the text, perhaps specify that the different conditions are characterized by a different combination of goal, effector, actuator and sensor.

Figure caption 3: perhaps point out in the caption that not all white targets in the Figure 3 were present on the screen at the same time. Otherwise this figure can easily be misinterpreted at first glance.

Figure caption 4: in the caption it says “representative participant trial for the Eye-Hand with Cursor condition” while all trials (=reaches) are depicted in the figure. That does not make sense to me.

Figure 4: it seems that there was an unequal number of fixations per each fixation target? Is that a result of randomization?

Reviewer #2: In the present study the authors evaluated the possibility of controlling an effector using eye movements. In particular, authors compared the accuracy of fixating a target in the presence and absence of an augmenting point-of-fixation feedback (cursor). Furthermore, the accuracy of the effector (cursor) end position was compared with that when a hand reach is initiated.

Overall, the manuscript is written in a rather clear manner in understandable language. However, there are several methodological issues that are in question and need to be improved from my point of view. In the following, I will comment on each of the sections separately.

Introduction

The motivation is described rather well. However, the section would benefit from a clearer description of how the effector is defined. Perhaps, authors can add a clarifying schematic or a diagram where the effector, actuator and sensor are indicated. In particular, how the 3 terms are defined in the context of an assistive system.

Furthermore, in the Discussion when comparing the results in the conditions without cursor and with cursor, authors interpret the better accuracy in case the cursor is present as a potential advantage of the gaze-point visual feedback. While I think it is a valid interpretation, I recommend dedicating few sentences with relevant references into the Introduction where the role of the gaze-point visual feedback is addressed.

In lines 52-63, authors mention “filtered unwanted eye movements…”. I recommend improving the phrasing here. From my understanding, in a reaching task there are different eye movements involved: fixations and saccades. In the present study you focus on fixations and the typical areas of interest fixated in a reaching task. Generally, saccades are also an important part of performing a reaching task, however, here what you do is applying a fixation detection algorithm.

Line 7: I believe the term “extremities” is constrained to hands and feet. As authors also include arm injuries, a broader term “limbs” would be more appropriate.

Line 99: ranged from -> ranged between

Materials and Methods

Participants information

The sample size in this study, 7 participants, is rather small. Was there an effect size analysis done before the study? While I understand that in the current global health situation it can be hard to recruit participants, I find it important to recognize this issue in the manuscript. Furthermore, please indicate whether participants were naïve or trained eye tracker users – from my experience it significantly affects the eye movement data.

Experimental conditions

Although I had to re-read the section few times, I believe the structure is logical. Also, if authors add a small diagram with effector, actuator, and sensor definition mentioned before, I think it would be easier to follow. I recommend, though, to align the order of the condition description with Fig.1: Either start the conditions description from “Eye-Alone” condition, or shuffle the conditions blocks in the figure such that it starts in the left upper corner with “Eye-Hand” condition. You can also add the listing numbers of the conditions to the figure to help the reader to follow.

From my perspective, it would also be helpful to add a sentence of reasoning for each condition. Specifically, compliment the conditions description with interpretation of each condition in the context of an assistive system. I could deduce the idea of comparing the cursor with the robotic arm only from the discussion, but not earlier.

Experimental setup and data acquisition

Generally, the main measured parameter in this study, the eye accuracy, is affected by the eye tracker accuracy. One major issue that I see in this study implementation is that there is no information about the distance between the eyes and the screen. Was it fixed? When it comes to accuracy measure in the context of eye trackers, it is typically given in degrees of visual angle (deg va), not meters. In other words, a fixed accuracy in deg va will result in a varied accuracy in meters depending of the distance from the screen. Did authors control for this during the experiment? What was the variability of the eye-screen distance during the experiment. I believe this is an important aspect for this study approach and it should be mentioned in the manuscript. In general, the manuscript would benefit from information regarding the variability of head rotation. Such, when it comes to fixation/saccade detection in case of free head movement, one common challenge is vestibulo-ocular reflex (eyes compensate the head movement in order to stabilize the image). Was it an issue in your data?

Also, was the height of the chair, table or screen adjusted for each participant such the forward gaze point would fall in the center of the screen? Please indicate this in the methods.

Analysis

If I understood correctly from the discussion, in the conditions where the cursor was present, the accuracy is defined as the error between the center of the target and the tip of the cursor. Is that right? Please indicate explicitly how you define accuracy for the cursor-present conditions as it comes in question whether you compare the same metric across conditions when comparing eye-only and the eye-cursor conditions.

Lines 200-201: The duration of the saccades is very much dependent on the task at hand and depending on the distance to the target quite often is longer (around 250 ms). But more importantly, I missed some more details on the fixation detection algorithm. I understand, the authors used velocity threshold algorithm for fixation identification. Which velocity threshold you applied? Which minimum duration is for it to be a fixation? Did you merge fixations that are very close in time into one larger fixation? How the fixation position is defined – is it a centroid of all raw gaze positions belonging to the fixation? Full understanding of the underlying parameters is important as it directly affects the results of the study, therefore, I recommend adding a more detailed description to the manuscript. Perhaps, this reference can be useful: (Salvucci & Goldberg, 2000)

Was the eye movement and hand movement data recording controlled from one device? Or did you have to additionally synchronize the clocks of both?

Results

Line 253-254: another reason is also a much higher sampling rate of the hand tracking device compared to the eye tracker

The figure 4A is very overloaded and therefore not very informative. Perhaps, authors could consider displaying only a small part of one trial. It is often helpful to plot the gaze data as a scatter plot, not a line plot. To indicate the order of the gaze positions a color coding can be used where a color bar would indicate time.

Line 262: I believe instead of “Hand-Alone” you meant “Eye-Hand”

Discussion

I rather enjoyed reading this section, it is easy to follow and I think the interpretation of the results is supported by the data. Authors nicely indicated the limitations of the study. The section can be complimented with other issues mentioned above (e.g. sample size).

Salvucci, D. D., & Goldberg, J. H. (2000). Identifying fixations and saccades in eye-tracking protocols. Proceedings of the Eye Tracking Research and Applications Symposium 2000, 71–78. https://doi.org/10.1145/355017.355028

6. PLOS authors have the option to publish the peer review history of their article (what does this mean?). If published, this will include your full peer review and any attached files.

Reviewer #1: No

Reviewer #2: No

---

## [Author Response · Author response to Decision Letter 0]

18 Nov 2021

All of these responses are included in the attached file "Response to Reviewers". These responses below summarize the main points.

Journal Requirements:

1) We reviewed the PLOS ONE style requirements and made a few changes, namely author affiliations and file naming.

2) We added a sentence in the Methods section that states that participants provided written informed consent. (p. 3, ll. 106-107)

3) We added the suggested sentence to the Cover Letter.

4) We added an in-text citation referring to the Supporting information in the Results section (p. 8, ll. 323). A caption for the Supporting Information was included in the previously submitted manuscript.

Reviewer 1:

1) Our approach here was to initially assess the data and provide a general description of the pattern of the results, and then follow up with the statistical results. However, we believe the reviewer is correct in pointing out the redundancies this approach introduces. We re-worked the section in question to eliminate the redundant report of the descriptive values and to suggest conclusions only after the statistical analysis had been reported (p. 8, ll. 324 – p. 10, ll. 382).

2) The reviewer makes a good point in that we are investigating the main effects here, not interaction terms. We addressed this comment by re-working the Results section as described in the previous comment response (p. 8, ll. 324 – p. 10, ll. 382).

3) We agree with the reviewer here that a non-significant t-test result does not allow us to accept the null hypothesis. Because we are trying to determine equivalence, rather than the difference between the samples, we should instead use an equivalence test with paired data. 

Here, we are comparing the mean eye fixation error in the ‘Eye-Alone with Cursor’ condition to the mean hand endpoint error in the ‘Eye-Hand’ condition. We want to see if the mean eye fixation error is equivalent to the mean hand endpoint error within an equivalence interval. If the difference between the mean eye fixation error and the mean hand endpoint error is within ±1cm of the mean hand endpoint error, we consider this to be equivalent. The equivalence test showed that the confidence interval was within the equivalence interval (p < .05), therefore we can claim that the means are equivalent. We have described all of this and updated the manuscript accordingly (p. 8, ll. 303-305 and p. 9, ll. 364-376).

4) Thanks to the reviewer for pointing out the typo. This has been corrected in the manuscript in the reworked section as part of the previous response (p. 9, ll. 364-376).

5) The reviewer makes a great point here about consistent terminology. Throughout the paper, we eliminate all references to “accuracy.” We instead only use the terms “Eye fixation error” and “Hand endpoint error”. Eye fixation error refers to the Euclidean distance between the center of the target and the participant’s point of gaze. Hand endpoint error refers to the Euclidean distance between the center of the target and the position of the motion capture marker at the end of the reaching movement. Additionally, this comment got us thinking about other terminology inconsistencies. We now only refer to the target which begins a reach as the “starting target”. The target that then appears on the screen that the participant looks at or reaches for is referred to as the “task target”. The starting target always appears in the same location (bottom-center of the screen). The task target appears randomly in one of the 6 determined positions. These terms are updated throughout the manuscript.

6) This is a good suggestion. We updated the figure caption in the manuscript.

7) The reviewer makes a good point here. We updated the figure caption to reflect the suggested change.

8) This is a good catch. In the manuscript, we defined one reach (movement from starting target, to task target, back to starting target) as one trial. We should have said “session” instead of “trial” in the figure caption. This has been updated in the manuscript. We also added a figure showing the initial movement in the horizontal direction of a typical trial. It highlights both the definition of a trial and the typical temporal sequence of events.

9) For this particular participant, it is true that there are fewer fixations in some of the targets. There are a few reasons for this. First, the starting target was displayed in every trial. Therefore, effectively there should be 6 times as many X’s within the starting target as there are in the task targets. Second, on occasion, the participant did not notice the appearance of a task target and did not make a movement toward it. We think this is due to the fact that participants relied on their peripheral vision to detect the appearance of the task target. Therefore, it makes sense that the outer targets, or targets more in the participants’ periphery, were the ones missed more often. Also, because the timing and locations of the targets were randomized, it was impossible for the participants to predict where and when the next target would appear. This also probably contributed to a few missed targets. These missed trials were not included in the results. We added details regarding these points to the manuscript (p. 11, ll. 450-455). We also changed the figure (now Figure 5) so that the starting target was a different color than the task targets. This should help to make the distinction between targets clearer.

Reviewer 2:

1) The reviewer makes a good suggestion here as to how to clarify the terminology. We modified the section in the Introduction to better describe the goal of the study (p. 3, ll. 79-80). We also added an example of how the 3 terms are defined in the context on an assistive system, as per the reviewer’s suggestions. We added this example in the Experimental Conditions sub-section of the Materials and Methods section (p. 4, ll. 123-131). 

2) This is a good suggestion. We added some information into the Introduction in response to this comment (p. 3, ll. 84-85).

3) We agree with the reviewer that the phrasing here is somewhat clunky. It is true that eye movements are characterized by both saccades and fixations, however, for this study, we were only interested in fixations, and not the eye movements leading up to them. The passage in question has been rephrased in the manuscript (p. 2, ll. 52-56).

4) This is a good catch. We have updated this in the manuscript (p. 1, ll. 7).

5) Updated (p. 3, ll. 104-105).

6) We agree that the sample size might be smaller than desired. We performed a power and sample size test. For the ANOVA comparison, to detect a difference of 1.3cm (the average difference between pairs of conditions), with a power of at least 90%, we would need 11 participants. Our current observed power for that measure was 75%. Similarly, for the equivalence test (see response to Reviewer 1, Comment 3), to detect a difference of 0.2cm (on average what we observed), with a power of at least 90%, we would need 8 participants. Our current observed power for that measure was 83%. So, while our power for each measure of this study was not bad, more subjects would have been better. This limitation has been included in the manuscript (p. 11, ll. 446-450).

We did not explicitly ask if the participants were familiar with other eye-tracking devices, however, none of the participants had used this system before and were all given the same instructions on how to use it. This has been added in the manuscript (p. 3, ll. 108-110).

7) We believe the reviewer makes a good suggestion here regarding the organization of this section. We ultimately decided to go with describing the experimental conditions in the following order: 1) Eye-Alone, 2) Eye-Hand, 3) Eye-Alone with Cursor, 4) Eye-Hand with Cursor (p. 4, ll. 115-116). To the figure (Fig 1), we added condition numbers which correspond to the order in which the conditions are listed in this section.

8) This is a good suggestion. We added a sentence or two to each condition providing some perspective on how each condition might be observed in the context of an assistive system (p. 4, ll. 151-153; p. 5, ll. 162-164; p. 5, ll. 175-176; p. 5, ll. 186-189).

9) The reviewer makes some good points regarding the factors which can lead to inaccuracies in eye-tracking. We do our best here to explain our thought process when designing this experiment with the primary goal being to understand if eye-movements are a good candidate input signal for controlling reaching motions with an assistive device. 

We had each participant sit in a chair facing the monitor. We positioned the chair a distance away from the monitor such that the participant could reach all of the target locations comfortably. The chair was not moved for the entirety of the testing session (all 4 experimental conditions). During data collection, we did not stabilize the head position. For this study, we were interested in the participant’s point-of-gaze on the screen. Our eye-tracker was a head-mounted system which contains a magnetic sensor for accounting for head orientation in point-of-gaze computation. In this case, we wanted the setup to resemble how an individual might control an assistive device with such a system, with the only constraint being the targets displayed in one plane (the monitor). Therefore, each participant should be able to maintain a steady point-of-gaze despite slight differences in head rotation. However, this is a potential source of error and has been added to the manuscript. 

With the setup we used, we believe that the distance between the participant and the monitor was fairly constant, within each data collection session. It is true that there may exist between-subject differences in this distance, but this was not measured in the present study. All of the above points have been addressed and added to the manuscript (p. 6, ll. 207-215 and p. 11, ll. 455-460).

10) The chair and table height were such that the participant’s forward gaze point fell approximately in line with the center of the screen, however this was not strictly controlled for. This element of the experimental setup was the same for each participant. This was clarified in the manuscript (p. 5, ll. 197-199).

11) Yes, this is correct. The only difference between the ‘Cursor’ and ‘No Cursor’ conditions is that the cursor is displayed so that the participant can see it. Thus, the difference is in providing visual feedback to the user. The computation of the fixation error is the same, regardless of whether the cursor is displayed. This has been made more explicit in the manuscript (p. 7, ll. 271-274).

Additional details about the cursor could have been included. We did not include a figure showing what specific form of visual feedback was presented to the user. This was an oversight on our part and has been addressed by adding a depiction of the cursor in the Target Array figure (Fig 3) and accompanying updated figure caption.

12) The fixation filter methods were based primarily on the methods outlined in the cited paper (Olsson, 2007). The steps were as follows:

1. Interpolate any missing data due to blinks or dropped frames

2. Compute the difference in the means between sliding windows. The typical length of fixations is 150ms – 600ms (Duchowski, 2017) (and rarely fewer than 100ms, according to the reviewer’s suggested reference (Salvucci & Goldberg, 2000)). Using these values and visually inspecting the data, we set our window size to 80ms. The window should be defined such that it is long enough to detect adequate changes in the signal, but not too long that it spans multiple fixations. 

3. Find the “peaks” within the difference vector. For that step, we used the MATLAB command findChangePoints() instead of the method described by Olsson.

4. Remove peaks that are too close together in the time domain. Because multiple saccades can occur within the specified averaging window, we only kept the peak with the highest magnitude within the averaging window.

5. The last step was to compute the spatial distance between fixations using the median and merge fixations which occur within a specified spatial radius (1.1cm in this case).

These steps are outlined in detail in the cited reference (Olsson, 2007), however we added more detail to the manuscript regarding the modifications we made and parameter values we used. We believe the reviewer makes a good point here that while we cite this paper in the manuscript, more details should be added in the text. This has been addressed in the manuscript (p. 7, ll. 245-265). 

Olsson P. Real-time and offline filters for eye tracking; 2007.

Duchowski AT. Eye tracking methodology: Theory and practice; 2017.

13) The eye data was recorded using the iSCAN etl-600 device and the hand data was recorded using the Optotrack motion capture system. These two signals were sent to a dSPACE real-time target computer which facilitated clock synchronization, experimental logic, and data logging. This was made clearer in the manuscript (p. 6, ll. 219-221).

14) This is a good point. We have added this to the manuscript (p. 8, ll. 319-321).

15) The primary purpose of this figure (now Fig 5) was to show how variable human eye movements can be compared to their hand movements, as well as visually demonstrate the performance of the fixation filter. However, we agree with the reviewer here that the figure is cluttered. We changed the figure in question to show the data from a different participant whose raw eye data was not as variable. We also did not plot as many trials to reduce the visual clutter a bit. In addition, we changed the color of the starting target to make the distinction between targets clearer.

To better show the temporal sequence of events, we added another figure (now Fig 4) showing the target, eye, and hand data in the horizontal direction for the first half of one representative trial. The purpose of this figure is to make clearer the definition of a trial (as per a comment from reviewer #1), as well as to highlight the temporal sequence of events. We believe the modification of the figure in question (Fig 5), and the addition of the one-dimensional single trial figure (Fig 4) will help to address both reviewers’ concerns and clarify the content for the audience.

16) This is a good catch. The type has been corrected in the manuscript.

17) We appreciate the kind words. We have taken to heart the comments and suggestions above and have re-worked the Discussion section to better reflect the changes we made in response to your previous suggestions and those of Reviewer 1.

---

## [Decision Letter · Decision Letter 1]

2 Dec 2021

PONE-D-21-26505R1Controlling an Effector with Eye Movements: The Effect of Entangled Sensory and Motor ResponsibilitiesPLOS ONE

Dear Dr. Schultz,

Thank you for submitting your manuscript to PLOS ONE. After careful consideration, we feel that it has merit but does not fully meet PLOS ONE’s publication criteria as it currently stands. Therefore, we invite you to submit a revised version of the manuscript that addresses the points raised during the review process.

The reviewers have been generally happy with your work. There is only one minor comment left from reviewer 1. Please address this in a minor revision.

We look forward to receiving your revised manuscript.

Kind regards,

Markus Lappe

Academic Editor

PLOS ONE

Reviewers' comments:

Reviewer's Responses to Questions

**Comments to the Author**

1. If the authors have adequately addressed your comments raised in a previous round of review and you feel that this manuscript is now acceptable for publication, you may indicate that here to bypass the “Comments to the Author” section, enter your conflict of interest statement in the “Confidential to Editor” section, and submit your "Accept" recommendation.

Reviewer #1: (No Response)

Reviewer #2: All comments have been addressed

2. Is the manuscript technically sound, and do the data support the conclusions?

Reviewer #1: Yes

Reviewer #2: Partly

3. Has the statistical analysis been performed appropriately and rigorously? 

Reviewer #1: Yes

Reviewer #2: Yes

4. Have the authors made all data underlying the findings in their manuscript fully available?

Reviewer #1: Yes

Reviewer #2: Yes

5. Is the manuscript presented in an intelligible fashion and written in standard English?

Reviewer #1: Yes

Reviewer #2: Yes

6. Review Comments to the Author

Reviewer #1: Most of my comments have been adequately addressed by the authors.

Regarding the equivalence test replacing the t-test, I miss the justification for the equivalence limits chosen. 1 cm might be a reasonable choice, but especially if the equivalence bounds are not set before the initial data analysis, the rationale behind the choice should be briefly explained in the manuscript.

P 11, ll. 436-437: typo: “despite” is repeated

Reviewer #2: The authors thoroughly addressed all of my comments. The manuscript significantly improved in its soundness and clarity. The sample size is rather small as was mentioned in previous comments, but I believe this issue was appropriately addressed in the limitations section and therefore should not confuse the reader when interpreting the conclusions of the paper.

7. PLOS authors have the option to publish the peer review history of their article (what does this mean?). If published, this will include your full peer review and any attached files.

Reviewer #1: No

Reviewer #2: No

---

## [Author Response · Author response to Decision Letter 1]

22 Dec 2021

1. Most of my comments have been adequately addressed by the authors.

Regarding the equivalence test replacing the t-test, I miss the justification for the equivalence limits chosen. 1 cm might be a reasonable choice, but especially if the equivalence bounds are not set before the initial data analysis, the rationale behind the choice should be briefly explained in the manuscript.

This is a good suggestion. We added a brief explanation regarding the choice of 1cm for the equivalence interval. To measure the hand motion, we placed a motion capture marker on a stylus, which was attached to the participant’s finger. During experimental conditions including hand motion, the participants were instructed to touch the tip of the stylus to the center of the target. The tip of the stylus had a diameter of 1cm. Therefore, if the eye fixation error was within 1cm of the hand error, we considered this to be equivalent, as reflected by the 1cm equivalence interval. We addressed this response in the manuscript (p. 9, ll. 332 - 335).

2. P 11, ll. 436-437: typo: “despite” is repeated

Good catch. This has been fixed (p. 10, ll. 396).

---

## [Editor Report · Decision Letter 2]

20 Jan 2022

Controlling an Effector with Eye Movements: The Effect of Entangled Sensory and Motor Responsibilities

PONE-D-21-26505R2

Dear Dr. Schultz,

We’re pleased to inform you that your manuscript has been judged scientifically suitable for publication and will be formally accepted for publication once it meets all outstanding technical requirements.

Kind regards,

Markus Lappe

Academic Editor

PLOS ONE
---

## [Editor Report · Acceptance letter]

25 Jan 2022

PONE-D-21-26505R2 

Controlling an effector with eye movements: The effect of entangled sensory and motor responsibilities 

Dear Dr. Schultz:

I'm pleased to inform you that your manuscript has been deemed suitable for publication in PLOS ONE. Congratulations! Your manuscript is now with our production department. 

Kind regards, 

on behalf of

Dr. Markus Lappe 

Academic Editor

PLOS ONE